Niche-related processes explain phylogenetic structure of acoustic bird communities in Mexico

Morán-Titla Christian D. 1
García-Chávez Juan-Hector 2
Lopez-Toledo Leonel 1
González Clementina clementina.gonzalez@umich.mx 1
1 Instituto de Investigaciones sobre los Recursos Naturales, Universidad Michoacana de San Nicolás de Hidalgo , Morelia , Michoacán , México
2 Facultad de Ciencias Biológicas, Benemérita Universidad Autónoma de Puebla , Puebla , Puebla , México
Manjarrez Javier
Electronic publication date: 2025 Jan 8
Publication date: 2025
Volume: 13
Electronic Location ID: e18412
Received 2024 Jun 4; Accepted 2024 Oct 7
Copyright: ©2025 Morán-Titla et al.
Copyright year: 2025
Copyright holder: Morán-Titla et al.
License: This is an open access article distributed under the terms of the Creative Commons Attribution License, which permits unrestricted use, distribution, reproduction and adaptation in any medium and for any purpose provided that it is properly attributed. For attribution, the original author(s), title, publication source (PeerJ) and either DOI or URL of the article must be cited.
License URL: https://creativecommons.org/licenses/by/4.0/

Keywords: Acoustic adaptation hypothesis, Bioacoustic indices, Environmental filters, Zapotitlán, Acoustic niche hyphothesis

Funding: The Coordinación de la Investigación Científica, Universidad Michoacana de San Nicolás de Hidalgo (2022) The Programa para el Desarrollo Profesional Docente to Clementina González The Consejo Nacional de Humanidades Ciencias y Tecnologías (CONAHCYT) graduate scholarship 764099 to Christian D. Morán Titla Funds were provided by a research grant from the Coordinación de la Investigación Científica, Universidad Michoacana de San Nicolás de Hidalgo (2022), and from the Programa para el Desarrollo Profesional Docente to Clementina González and by a graduate scholarship (764099) to Christian D. Morán-Titla from the Consejo Nacional de Humanidades Ciencias y Tecnologías (CONAHCYT). The funders had no role in study design, data collection and analysis, decision to publish, or preparation of the manuscript.

==============================
Acoustic communities are acoustically active species aggregations within a habitat, where vocal interactions between species can interfere their communication. The acoustic adaptation hypothesis (AAH) explains how the habitat favors the transmission of acoustic signals. To understand how bird acoustic communities are structured, we tested the effect of habitat structure on the phylogenetic structure, and on the phylogenetic and vocal diversity of acoustic communities in a semi-arid zone of Mexico. From autonomous recordings in three types of vegetation (crop fields, tetecheras, and mesquiteras), which differ in terms of complexity and canopy openness, we evaluated sound attenuation, and estimated metrics of phylogenetic structure and diversity as well as acoustic diversity with the use of two indices. Mesquiteras showed greater vegetation density, more attenuation, more vocal diversity, as well as a phylogenetic structure that tended towards overdispersion, in contrast to crop fields that showed less vegetation density, less attenuation, less vocal diversity and more phylogenetic relatedness, while tetecheras showed intermediate patterns. Phylogenetic structure was explained by vegetation density and excess attenuation. The higher vocal diversity, phylogenetic structure tended towards overdispersion. These results suggest a role for environmental filters in the crop fields, where more closely related species with similar vocal characteristics coexist (supporting AAH), and probably competitive exclusion in the mesquiteras, where more distantly related species coexist, promoting vocal diversity. This study offers information about the influence of habitat on the acoustic community structure, which could inform our understanding of the distribution of species from acoustic perspective.

Introduction

Acoustic signals in birds have important functions in both intra and interspecific communication, such as resource defense, mate attraction, and recognition, among others (Catchpole & Slater, 2008; Bradbury & Vehrencam, 2011). For this reason, effective communication in these contexts is essential for survival and reproduction (Yasukawa, 1989; Catchpole & Slater, 2008). Although many bird species are capable of learning their vocalizations (i.e., songbirds, parrots, and hummingbirds), vocal characteristics of the species maintain an important genetic basis; therefore, similarities between species depend, in large part, on their phylogenetic relationships (Kroodsma & Candy, 1985; Payne, 1986; Price & Lanyon, 2002). However, vocal characteristics of species can be shaped by the physical properties of the habitat, and can converge across species (Seddon, 2005; Nicholls & Goldizen, 2006). For example, vocalizations of distantly related species that live in similar habitats may be more similar than those of closely related species that live in different habitats (Badyaev & Leaf, 1997; Tobias et al., 2010). Thus, the acoustic adaptation hypothesis (AAH) tries to explain the evolution of acoustic signals depending on the structure of the habitat (Morton, 1975). According to this hypothesis, birds adapt their vocalizations to local conditions (Morton, 1975; Hansen, 1979), under the assumption that the signals are directionally selected, improving their transmission and minimizing their attenuation and degradation (Morton, 1975; Dabelsteen, Larsen & Pedersen, 1993; Boncoraglio & Saino, 2007; Ey & Fischer, 2009).

Attenuation is a measure of energy loss of sound propagation that depends on distance as well as on habitat structure and climatic factors. Excess attenuation, defined as observed attenuation in addition to that expected from spherical propagation, depends on habitat and climatic conditions, but also depends on the level of the frequency of sounds (Catchpole & Slater, 2008). Vegetation foliage affects sound transmission showing greater attenuation in denser vegetation; however, the effect depends on the frequency, since high frequency sounds (greater than 4 kHz) tend to show greater excess attenuation, while those below this threshold tend to be amplified (Martens, 1980). Degradation or distortion is especially problematic in dense vegetation covers such as forests, since reflection off the ground or canopy, as well as echoes off tree trunks, can make it very difficult to distinguish or detect successive acoustic features or the shape of individual elements, mainly affecting fast frequency modulation signals (Catchpole & Slater, 2008). Thus, in closed and dense habitats, such as forests, which have different vegetation strata that absorb acoustic energy (Martens & Michelson, 1981), sound tends to be attenuated and degraded due to reverberations (Marten & Marler, 1977; Richards & Wiley, 1980; Boncoraglio & Saino, 2007). These effects are intensified with increasing frequency, bandwidth, and rapid frequency modulations (Naguib, 2003). Therefore, under these conditions, low-frequency signals with simpler structure (i.e., pure tones or low modulation elements) and slower song rate will be better transmitted in closed, densely foliated habitats (Morton, 1975; Slabbekoorn & Smith, 2002; Kirschel et al., 2009). However, dense and more heterogeneous habitats with different vegetation strata will also allow different birds species to sing at different height from the ground to the canopy, favoring different sounds that are optimally transmitted (Nemeth, Winkler & Dabelsteen, 2002). In contrast, open homogeneous habitats, such as grasslands or crop fields, have fewer vegetation strata that reflect or absorb acoustic energy, however, factors such as wind or temperature can increase the effects of attenuation (Morton, 1975). Under these conditions, repeated signals with high and fast frequency modulations and wide bandwidths will be better transmitted (Naguib, 2003; Derryberry, 2009).

An acoustic community is defined as a group of species established in a habitat that are acoustically active at different times of the day or night, and in different seasons of the year (Malavasi & Farina, 2013; Farina, Pieretti & Malavasi, 2014). Within an acoustic community, simultaneous vocalizations emitted by individuals of different species with similar acoustic characteristics are more likely to be masked, interfering with the effectiveness of intra and interspecific communication (Wong, Parada & Narins, 2009; Hart et al., 2015). In response to masking and interference of acoustic signals, selection can favor species of a community to occupy different regions of the spectral and temporal parameters in the acoustic space that minimize overlap (Schmid, Römer & Riede, 2013; Krishnan & Tamma, 2016). That is, the signal or acoustic space is divided into frequencies and time where the signals disperse (Luther, 2008; Cardoso & Price, 2010), reducing overlapping (Krause, 1993; Farina et al., 2011). This is known as the acoustic niche hypothesis (ANH), which proposes that competition between species for the acoustic space has promoted the diversification of acoustic signals, resulting in sounds produced at different temporal and spectral intervals (Krause, 1993).

To understand how acoustic communities are structured, it is necessary to incorporate phylogenetic information as an important component for understanding the structure and assembly of acoustic communities (Chhaya et al., 2021). Because phylogenetic trees are representations of the accumulation of biological differences along the branches of a certain taxon, they can be used to describe, explain, or predict ecological and evolutionary processes involved in species assembly into biological communities (Tucker et al., 2017). In accordance with this idea, several metrics have been developed to assess patterns of community structure and diversity from a phylogenetic perspective (Vellend et al., 2011; Cadotte & Davies, 2016; Tucker et al., 2017). Phylogenetic structure refers to the nonrandom distribution of species within a community with respect to phylogeny (Webb et al., 2002), while phylogenetic diversity is a measure of biodiversity that incorporates phylogenetic differences between species (Faith, 1992). By estimating how closely related a pair of species are on average within a community, three possible patterns can be obtained: clustering, overdispersion, or a random pattern. Phylogenetic clustering occurs when closely related species share similar conditions and resources as a consequence of tolerance to the environment (environmental filtering; Webb et al., 2002). In contrast, phylogenetic overdispersion of a community suggests competition by causing overdispersion of conserved traits, but may also be caused by environmental filters when distantly related species converge for important ecological traits (Webb et al., 2002). Therefore, under the niche assembly theory, phylogenetic structure metrics can be used as a proxy of how similar a pair of species are in certain traits, assuming close relatives are more similar to each other than more distantly related species (Cavender-Bares et al., 2009). Under a neutral perspective, a random phylogenetic pattern is expected since the species are considered ecologically equivalent (Hubbell, 2001; Kembel & Hubbell, 2006).

In this study, we evaluated the effect of habitat structure (acoustic adaptation hypothesis), on the phylogenetic structure and phylogenetic and vocal diversity of acoustic bird communities in a semi-arid zone of central Mexico. Our aim is to understand how communities are structured from the acoustic perspective, and infer the processes involved. Accordingly, we propose the following hypothesis: Habitat influences patterns of phylogenetic structure and diversity as well as patterns of vocal diversity of the acoustic bird community. Under this hypothesis we formulated the following predictions: (A) we expect significant differences in excess attenuation, phylogenetic structure, phylogenetic and vocal diversity among habitat types; (B) we expect a negative relationship between phylogenetic structure and vegetation structure, and between phylogenetic structure and excess attenuation, and a positive relationship considering phylogenetic diversity; and (C) we expect a negative relationship between phylogenetic structure and acoustic diversity, where the less closely related the species that coexist (overdispersed) the higher the acoustic diversity. These predictions are based on the idea that more distantly related species are expected to coexist in denser habitats with more niches available which would imply the existence of more diverse vocal features that could be effectively transmitted from different vegetation strata. On the other hand, more open habitats would favor less diverse sounds shared by more closely related species as a consequence of tolerance to harsher environments. We used phylogenetic information as a proxy for vocal differences between species, assuming that close relatives will tend to have more similar vocal characteristics than more distantly related species (Cavender-Bares, et al. 2009).

Materials & Methods

Study site

The study was conducted in the Zapotitlán Valley, a semi-arid region of the Tehuacán-Cuicatlán Biosphere Reserve in central Mexico, with the permission of the authorities of the municipality and the Helia Bravo Hollis Botanical Garden. The reserve is located southeast of the state of Puebla comprising an area of 10,000 km2 between 17°48′ and 18°58′ North latitude and 96°40′ and 97°43′ West longitude. The region is characterized by two short rainy seasons from April to May and from September to October with a total annual rainfall of ∼412 mm and an annual mean temperature between 17.6 and 23.7 °C. Xeric scrubland characterized the vegetation of the region and includes three main plant associations: (1) Tetechera, habitat dominated by the columnar cactus Neobuxbaumia tetetzo, accompanied by Mimosa luisana, Aeschynomene compacta, Cordia cylindrostachya, Mammillaria colina, Ruellia sp., Eysenhardtia polystachia, and Opuntia pilifera; (2) Mesquitera, habitat dominated by Prosopis laevigata accompanied by Parkinsonia praecox, Myrtillocactus geometrizans, Stenocereus pruinosus, S. stellatus, and Pachycereus hollianus; and (3) Sotolinera, habitat dominated by Beaucarnea gracilis accompanied by Yucca periculosa, Myrtillocactus geometrizans, Pachycereus hollianus, Opuntia pilifera, and Agave kerchovei (Rzedowski, 1978). Crop fields, most of them abandoned, with secondary vegetation scattered on plot edges can also be found at the study site. In the region, 108 species of birds belonging to 13 orders and 34 families have been registered, with nine endemic, 30 migratory, 22 local migratory, and 56 resident species (Camacho-Morales, 2001).

Field procedures

We selected three types of habitat showing contrasting patterns of vegetation: (1) abandoned crop fields with secondary vegetation (open habitat); (2) tetechera, dominated by columnar cacti (intermediate in canopy openness); and (3) mesquitera (more closed and denser habitat; Fig. 1). We delimited five spatial blocks separated by at least 700 m, each one comprising the three types of habitat separated by at least 80 m from each other, with a total of 15 plots (Fig. 2). To obtain an estimate of vegetation structure, as a proxy for habitat density and heterogeneity, between March and April 2021, we set four transects of 25 m long by 2 m wide on each of the 15 plots, and registered each individual of every single plant species. Plant species were identified, in some cases with the aid of a local field guide. From each individual taller than 1.5 m, we measured the canopy diameter and basal cover (sum of the diameters at breast height of each branch). We also measured the canopy opening every 5 m within the transect, with a concave spherical densiometer placed at chest height. With these variables we performed a principal components analysis (PCA), and from the correlation matrix we extracted the factor scores of PC 1, which explained 68.08% of the variation. The lowest values (negative numbers) of the factor scores were associated with the highest values of vegetation measures, which implies greater density and coverage. These values were multiplied by -1 to have a consistent scale with density, that is, higher values imply higher vegetation density and coverage.

Figure 1 Three habitat types in the Zapotitlán Valley whitin the Tehuacán-Cuicatlán Biosphere Reserve in central Mexico included in this research.

(A) Mesquitera, (B) tetechera, and (C) crop fields. Photographs taken by Christian D. Morán-Titla (A and B) and by Lizzette Morán-Titla (C).

Figure 2 Study site and spatial distribution of the sampling sites.

Five blocks (ellipses), including the three types of habitat (sampling plots) and three sampling points within each plot, were delimited. Jardín = Helia Bravo Hollis Botanical Garden; Barragán = plots of Mr. Juan Barragán Rivera; Vicente = plots of Mr. Vicente Mendoza; Tilapa = plots of Mr. Joaquín Pacheco; Sotolinera = plots of Mr. Daniel Pacheco.

As another measure of habitat density and heterogeneity but in acoustic terms, we evaluated sound attenuation in each of the 15 plots studied. To estimate sound attenuation, we generated artificial tones with frequencies from 0.2 to 5 kHz (frequencies at which birds are more sensitive to perceive the sound; Dooling, 1982), with a frequency range of 0.5 kHz. The audio file of 1 min included three repetitions of each tone separated by 0.1 s. We played the audio from a SAMSUNG cell phone connected to a BOSE speaker placed at chest height with an output of 80 dB at each point where an Audiomoth recorder was placed. We recorded the playback at 1 m (reference distance) and 50 m (average distance at which a bird can be heard in closed vegetation), with a TASCAM DR 100 MK II recorder and a Sennheiser MKE 600 shotgun microphone. From these recordings we estimated the excess attenuation (Morton, 1975; Marten & Marler, 1977) in the baRulho package (Araya-Salas, 2020) for R v.4.1.2 (R Core Team, 2021). The excess attenuation measures the loss of 6 dB of signal amplitude every time the emission distance is duplicated, where low values indicate little attenuation (Catchpole & Slater, 2008). Estimations were averaged per plot (n = 15).

During the same period, which comprises the peak in bird breeding season, we placed three autonomous Audiomoth recorders (Open Acoustic Devices) within each habitat of each block, that is, nine recorders per block per day (Fig. 2). In total, six repetitions were performed in each block, considering as a repetition one day of recording within a block. Each repetition was conducted every five days. Recorders were separated by at least 50 m from each other, programmed to record simultaneously for 2 h per day divided into 4 periods of half an hour (6:30–7:00; 7:30–8:00; 8:30–9:00; and 9:30–10:00 hrs). Recordings were performed in WAV format, at a sampling rate of 32 kHz with the maximum recording gain. Due to the high consumption of computational resources that recordings processing and the estimation of bioacoustic indices requires, each recording of half an hour was segmented into six audios of 5 min. All singing species constantly recorded with the autonomous recorders were identified and used to construct a presence/absence matrix for each plot. Only one species was recorded once, and was not considered for the analyses.

Acoustic diversity through the calculation of acoustic indices

By using the soundecology package (Villanueva-Rivera & Pijanowski, 2018) for R, we obtained: (i) the Bioacoustic Index (BI; Boelman et al., 2007), and (ii) the Acoustic Complexity Index (ACI; Pieretti, Farina & Morri, 2011) for the 6480 recordings obtained, considering a band filter below 0.2 and above 11 kHz in order to avoid frequency bands derived from anthropophonies and abiophonies. The Bioacoustic Index estimates the area under the curve that includes all the frequency bands associated with the highest values of amplitude (measured in dB), therefore, it is a function of both the sound level (amplitude) and the number of frequency bands emitted by the avifauna (Boelman et al., 2007). The more frequency bands recorded and with different dB levels, the higher the index values, which indicates more acoustic diversity (Boelman et al., 2007). The Acoustic Complexity Index is based on biotic sounds characterized by the variability of signals with different amplitudes (different intensity peaks) in each time segment of the recording (Pieretti, Farina & Morri, 2011; Farina & Morri, 2008). The signals in the audio do not maintain a single amplitude peak, which would indicate the existence of several individuals singing over time, and therefore higher values of this index indicate more acoustic diversity (Pieretti, Farina & Morri, 2011). In summary, the BI estimates acoustic diversity in terms of frequency bands and the ACI does it in terms of time. Finally, index values corresponding to recordings of the hours with the highest activity were selected (7:30–8:00) and averaged per plot (n = 15) for further analyses.

Phylogenetic diversity and structure estimates

From the bird species recorded by Camacho-Morales (2001) within the study area in addition to the species that we recorded in the field during this study, we obtained a phylogenetic subgroup from the BirdTree platform (https://birdtree.org) derived from the mega phylogeny of 9,993 extant bird species constructed by Jetz et al. (2012). We subsampled 1000 “stage 2” trees (Erickson), and with consensus function of ape package for R, we obtained a majority-rule consensus tree. Using the consensus tree and the presence/absence matrices of species recorded from the autonomous recordings, we calculated measures of diversity and phylogenetic structure of each plot (n = 15). The phylogenetic diversity index (PD) proposed by Faith (1992) quantifies the total branch length spanned by the phylogenetic tree including all the species in a community, therefore is an estimation of the evolutionary history of the community. This measure then explicitly incorporates differences between species and not just richness (Faith, 1992). Two measures of phylogenetic structure were also estimated: the net relatedness index (NRI) and the nearest taxon index (NTI) (Webb et al., 2002; Webb, Ackerly & Kembel, 2008). NRI quantifies the standardized effect size of the mean phylogenetic distance between all species pairs in the community, while the NTI quantifies the mean phylogenetic distance to the nearest taxon (Webb et al., 2002). Positive values of NRI and NTI indicate that species within a community are more closely related than would be expected by chance (phylogenetic clustering), while negative values indicate that species within a community are less closely related than would be expected by chance (phylogenetic overdispersion). NRI measures the degree of relatedness between all the members of the community, emphasizing deeper divergences, while the NTI is more sensitive towards the tips of the phylogeny. Finally, to determine whether communities showed phylogenetic clustering or overdispersion patterns, we compared the observed values of NRI and NTI to the patterns expected under null community structure, using taxa.levels as a model to generate 1000 null communities. We estimated both the indices and the comparisons with null communities with the picante package for R (Kembel et al., 2010).

Statistical analyses

Most of the birds registered in our study plots were defending territories during the censuses, since most species were constantly registered with the autonomous recorders. We acknowledge that the movement of some species between plots may affect our results. One way to determine the spatial independence of the sampling points is controlling for the effect of geographical distance using the Moran’s I test. Results showed that none of the variables were associated with geographic distance between sampling points (vegetation (PC1), P = 0.25; excess attenuation, P = 0.71; BI, P = 0.12; ACI, P = 0.57; PD, P = 0.061; NRI, P = 0.49; NTI, P = 0.93), therefore, we did not consider the spatial effect in our models.

To test for differences in vegetation structure, excess attenuation, structure and phylogenetic diversity indices as well as in acoustic diversity among habitat types (prediction A), we performed ANOVAs, since all variables, except NRI, met the assumption of normality of residuals (Table S1). For NRI, we performed a generalized linear model with Gamma error distribution. Since the Gamma distribution assumptions imply positive non zeros values, we added a constant to the NRI to avoid negative values and zeros preserving the variance. In the models, habitat type was the fixed factor and vegetation structure (scores of PC1), excess attenuation, PD, NRI, NTI, BI and ACI the dependent variables in separate models. We included a model with vegetation structure to verify that this measure is a good predictor of habitat density, considering that higher values of the factor scores indicated a higher vegetation density.

To test the relationship between habitat characteristics (vegetation structure and excess attenuation) and phylogenetic structure and diversity (prediction B), we performed multiple linear regressions between PD, NRI and NTI as a function of vegetation structure (scores of PC1), and excess attenuation. Finally, to test the relationship between phylogenetic structure and acoustic diversity (prediction C), we performed multiple linear regressions between NRI and NTI and acoustic indices (BI and ACI). For each regression the null hypothesis of residuals normality was not rejected.

Results

Vegetation diversity and structure

The mesquitera (Fig. 1A) had the highest diversity of plant species (26 species) with six different life forms, such as thorny shrubs, herbaceous, cacti (globose, columnar, creeping and climbing), epiphytes, medium-sized trees and agaves. Therefore, it was the most heterogeneous habitat since included more plant species with different strata and life forms. It was also the habitat with the highest basal vegetation cover, the highest percentage of canopy cover, and the largest canopy diameter, showing higher vegetation density. In contrast, crop fields were the least dense, most open habitat, with lower basal cover, lower leaf density, bare soil and scattered shrubs arranged on the plot edges, showing less complex vegetation (Fig. 1C). We recorded fewer plant species (16 species) and fewer life forms in this type of habitat. We only registered scattered herbaceous plants, some columnar cacti, and some thorny shrubs. This habitat was more homogeneous as fewer plant species and fewer life forms were recorded. In the tetechera, we recorded 21 plant species and it was dominated by columnar and globose cacti, although thorny bushes, agaves, bromeliads, small trees, and in some areas elephant-foot tree (Beaucarnea gracilis) could be found (Fig. 1B). Tetecheras showed an intermediate vegetation density pattern compared to crop fields and mesquiteras, according to its characteristics such as a moderately closed habitat, low leaf density but high basal cover, and showed a more homogeneous vegetation structure compared to the mesquitera.

Vegetation structure measures were a good predictor of habitat density since the factor scores of PC1 showed significant differences between habitat types with 79.81% of the variation explained (Table 1). Mesquiteras showed the highest values of the factor scores of PC1, indicating a higher vegetation density, followed by tetecheras, and finally crop fields that showed the lowest values of the factor scores, and therefore lower vegetation density (Fig. 3A).

Table 1 ANOVA results for vegetation structure, excess attenuation, phylogenetic indices and acoustic indices depending on the habitat type (crop fields, tetechera and mesquitera) in separate models.

Fixed factor	Response variables	df	F	P	Explained variation (SMCG/SMCT)	
Habitat type	Vegetation (PC1)	2	23.72	0.0001	0.7981	
	Excess attenuation	2	6.73	0.0109	0.529	
	PD	2	4.68	0.0314	0.4382	
	NRI	2	27.46	0.0001	0.8207	
	NTI	2	10.10	0.0027	0.6275	
	BI	2	7.08	0.0093	0.5412	
	ACI	2	6.28	0.0136	0.5114	
Notes.

PD phylogenetic diversity index

NRI Net relatedness index

NTI nearest taxon index

BI bioacoustic index

ACI acoustic complexity index

SMCG SMCG is the group sum of squares and SMCT is the total sum of squares

Figure 3 Violin plots showing the distribution of (A) vegetation structure and (B) excess attenuation, by habitat type (crop fields, tetechera and mesquitera).

The red cross represent the median. Letters above the plots represent differences between groups according to the Tukey HSD test.

Excess attenuation, phylogenetic structure, phylogenetic and vocal diversity as a function of habitat (prediction A)

Excess attenuation was significantly different between habitat types, explained by 52.90% of the variation (Table 1). Mesquiteras showed the highest excess attenuation, while crop fields and tetecheras showed the same lower excess attenuation (Fig. 3B). This result indicated that the habitat influences patterns of sound intensity loss, where the habitat with higher attenuation is denser or more complex.

Phylogenetic diversity (PD) was significantly different between habitat types with an explained variation of 43.82% (Table 1). PD values were lower in crop fields compared to tetecheras and mesquiteras, however significant differences were only detected between crop fields and tetecheras (Fig. 4A). NRI and NTI showed positive and significantly higher values in the crop fields (more phylogenetically clustered), while tetecheras and mesquiteras showed non-significantly different negative values in NRI (more phylogenetically overdispersed) and positive in NTI (Figs. 4B, 4C). Explained variation was 82.07% and 62.75% respectively (Table 1). Four of the five crop field plots showed a phylogenetic clustering pattern relative to null communities. In tetecheras and mesquiteras, values were not significantly different from null communities.

Figure 4 Violin plots showing the distribution of (A) phylogenetic diversity index (PD), (B) net relatedness index (NRI) and (C) nearest taxon index (NTI) by habitat type (crop fields, tetechera and mesquitera).

The red cross represent the median. Letters above the plots represent differences between groups according to the Tukey HSD test.

Bioacoustic (BI) and acoustic complexity (ACI) indices, also differed between habitat types, with 54.12% and 51.14% of variation explained respectively (Table 1). Mesquiteras showed the highest values of BI while crop fields and tetecheras did not show differences (Fig. 5A), therefore, mesquiteras showed more acoustic diversity in terms of frequency. Regarding ACI, mesquiteras showed significantly higher values than tetecheras but not than crop fields (Fig. 5B). This result indicated that more diverse acoustic signals over time (during the course of the recording) was shown in mesquiteras compared to tetecheras.

Figure 5 Violin plots showing the distribution of (A) bioacoustic index (BI) and (B) acoustic complexity index (ACI) by habitat type (crop fields, tetechera and mesquitera).

The red cross represents the median. Letters above the plots represent differences between groups according to the Tukey HSD test.

Relationships between phylogenetic structure/diversity and properties of the habitat (prediction B)

Net relatedness index (NRI) was negatively related to vegetation structure and positively related to excess attenuation. NRI explained 27% of the variability in vegetation structure, and 27% in excess attenuation (Table 2). The interaction vegetation structure × excess attenuation was also significant, explained by 16% (Table 2). The significance of this interaction was influenced by multicollinearity among variables since vegetation structure and excess attenuation are strongly correlated (ρ = 0.71, P < 0.01). Due to the confusing trends of the multiple regression, we ran simple regressions and both were negative (vegetation structure, t = −3.99, P = 0.001; excess attenuation, t = −2.22, P = 0.04). Nearest taxon index (NTI) was only negatively related to excess attenuation, explained by 31% (Table 2). The results indicate that the higher the vegetation density and the higher excess attenuation of the habitat, the phylogenetic structure tends towards overdispersion. Phylogenetic diversity was not related to vegetation structure or excess attenuation (Table 2).

Table 2 Results from multiple linear regressions of the structure and phylogenetic diversity indices as a function of vegetation structure and excess attenuation.

	Response variables	m	t	P	r2	P (residuals)	
NRI	Vegetation structure	−4.67	−2.96	0.0012	0.27	0.6037	
	Excess attenuation	0.98	0.33	0.0011	0.27		
	Vegetation × attenuation	3.08	2.47	0.0131	0.16		
							
NTI	Vegetation structure	−1.39	−0.98	0.0738	–	0.6134	
	Excess attenuation	−1.15	−0.43	0.0106	0.31		
	Vegetation × attenuation	0.79	0.71	0.4779	–		
							
PD	Vegetation structure	723.9	1.20	0.2889	–	0.0945	
	Excess attenuation	−194.9	−0.17	0.3121	–		
	Vegetation × attenuation	−496.5	−1.04	0.2953	–		
Notes.

NRI Net relatedness index

NTI nearest taxon index

PD phylogenetic diversity index

m value of the slope

t test statistic

P probability value

r2 coefficient of variation explained; P (residuals), probability value obtained from the normality test (Shapiro–Wilk) of the model residuals

Relationships between phylogenetic structure and acoustic diversity (prediction C)

NRI was negatively related to the Bioacoustic Index (BI), and positively related to the Acoustic Complexity Index (ACI). NRI explained 28% of the variability in BI, and 32% in ACI (Table 3). NTI was negatively related only to BI, explained by 23%. None of the interactions were significant. These results indicate that the less closely related the species that coexist (phylogenetic structure towards overdispersion) the higher the BI, and the more closely related the species that coexist (phylogenetic structure tend towards clustering), the higher the ACI.

Table 3 Results from multiple linear regressions of the phylogenetic structure indices as a function of the bioacoustic indices.

	Response variables	m	t	P	r2	P (residuals)	
NRI	BI	−18.75	−0.66	0.0052	0.28	0.792	
	ACI	0.003	0.08	0.0028	0.32		
	BI × ACI	0.002	0.61	0.5388	–		
							
NTI	BI	−27.64	−1.175	0.0384	0.23	0.4061	
	ACI	−0.028	−0.819	0.1982	–		
	BI × ACI	0.004	1.147	0.2514	–		
Notes.

NRI Net relatedness index

NTI nearest taxon index

BI bioacoustic index

ACI acoustic complexity index

m value of the slope

t test statistic

P probability value

r2 coefficient of variation explained; P (residuals), probability value obtained from the normality test (Shapiro–Wilk) of the model residuals

Discussion

Consistent with the goals of this study, we found an effect of niche-related processes on the structure of acoustic communities. In general, we found that the habitat influences patterns of phylogenetic structure and diversity as well as patterns of vocal diversity of the acoustic bird community. Our data specifically indicated that: (1) in denser habitats, with higher excess attenuation, more distantly related species coexist (phylogenetic structure towards overdispersion) that showed higher acoustic diversity compared to less dense habitats. (2) The higher the vegetation density and the higher excess attenuation of the habitat, less closely related species coexist. (3) The less closely related coexisting species, the higher the acoustic diversity in terms of frequency, but the more closely related coexisting species, the higher the acoustic diversity in terms of time.

Habitat structure has been considered one of the most important selective forces shaping the evolution of bird song, allowing individuals to communicate more efficiently (i.e., acoustic adaptation hypothesis, AAH). This hypothesis has been supported with a lot of evidence, but also has shown mixed results (reviewed in Boncoraglio & Saino, 2007; Ey & Fischer, 2009). This lack of support is due, in part, to the fact that the AAH ignores the potential role of energetic costs of singing and selection by eavesdroppers (Boncoraglio & Saino, 2007). In general, our results support the AAH, suggesting that in more dense, complex and heterogeneous habitats, where different vegetation strata offer more niches, allowing different bird species to sing at different heigh from the ground to the canopy, the acoustic signals tend to be diverse. In other words, vegetation density and heterogeneity can induce different sound propagation environments that favor the transmission of different ranges of acoustic signals (Nemeth, Winkler & Dabelsteen, 2002; Rodriguez et al., 2014; Chitnis, Rajan & Krishnan, 2020; García-Navas, Feliu & Blumstein, 2023). This is reflected in the coexistence of more distantly related species (i.e., greater phylogenetic overdispersion). In contrast, more homogeneous and open habitats, such as crop fields, maintain low levels of excess attenuation, but the available niches and the possibility to sing from different heights are also reduced. As a consequence, closely related species (i.e., phylogenetic clustering shown by NRI and NTI), that share acoustic features that may be better transmitted in such open conditions, coexist.

According to vegetation density and excess attenuation measurements, if the habitat is dense and heterogeneous with high excess attenuation as the mesquitera, different features of the acoustic signals will be favored. Higher excess attenuation in the mesquitera, compared to the other habitats, is probably the result of greater foliar density as well as the presence of understory and woody plants distributed in different strata that absorb sound (Dabelsteen, Larsen & Pedersen, 1993; Catchpole & Slater, 2008). In the understory, where the vegetation is denser, the best transmitted songs are those with low frequencies with little modulation (i.e., pure tones), emitted at slower rates (Morton, 1975; Slabbekoorn & Smith, 2002; Kirschel et al., 2009). While higher frequency songs with different modulation rates will be emitted towards the top of the trees. The acoustically active species that we found most frequently in the mesquitera were: Columbina inca, Columbina passerina, Leptotila verreauxi, Zenaida macroura, Zenaida asiatica, (Columbidae) and Momotus mexicanus (Momotidae), species with low frequency songs (0.05–0.3 kHz) and pure tones; Nyctidromus albicolis (Caprimulgidae), Phainopepla nitens (Ptilogonatidae), and Icterus pustulatus (Icteridae), species with intermediate frequency songs (2-5 kHz) and slight frequency modulations; Micrathene whitneyi (Strigidae), Empidonax traillii (Tyrannidae) and Catherpes mexicanus (Troglodytidae), species with higher frequencies songs (6–10 kHz) and with faster frequency modulations. These species emit songs that ranges from very low to very high frequencies and belong to several bird families. The occurrence of these species in the mesquitera explains the higher BI (greater number of frequency bands with different dB levels), since phylogenetic relatedness is low, and thus, there is more acoustic diversity in terms of frequency, however in terms of time (ACI) is similar to the crop fields.

On the other hand, in an open and homogeneous habitat such as the crop fields, with increased wind currents, higher temperature due to greater solar radiation, and the absence of obstacles such as resonance boxes that prevent the reflection of sound (less excess attenuation), high frequency songs with short but repeated elements with rapid frequency modulations were favored. These are characteristics of songs that are generally transmitted over long distances. The most frequently species found in the crop fields were: Peucaea mystacalis, Spizella passerina (Passerellidae), Molothrus aeneus (Icteridae), Passerina caerulea (Cardinalidae), Melozone fusca (Emberizidae), Pyrocephalus rubinus, and Tyrannus vociferans (Tyrannidae). These species emit songs of much higher frequencies and of wider bandwidths (3–10 kHz). The structure of their vocalizations is very similar, such as the presence of short and repeated elements, rapid frequency modulations and short duration elements. The occurrence of the species in the crop fields explains the lower number of frequency bands with different dB levels (lower BI).

Finally, in the tetechera, the density and heterogeneity of the vegetation was intermediate and its excess attenuation levels were like those of the crop fields, so it tended to show bioacoustic patterns similar to this habitat. However, the phylogenetic patterns were similar to those of the mesquitera. The most frequently found species in the tetechera were: Toxostoma curvirostre, Mimus polyglottos (Mimidae), Melanerpes hypoppolius, Melanerpes aurifrons (Picidae), Campylorhynchus jocosus, Catherpes mexicanus, Thryomanes bewickii, Salpinctes obsoletus (Troglodytidae), Icterus pustulatus, Dives dives, Molothrus aeneus (Icteridae), Pheucticus melanocephalus, Pheucticus chrysopeplus (Cardinalidae), and Nyctidromus albicollis (Caprimulgidae). Most of these species are characterized by long and complex songs, composed of short but diverse elements with modulated frequencies that ranges from 1.5 to 7.5 kHz.

Environmental filters are defined as those environmental or habitat characteristics that condition the conservation of species traits in a community (Duarte, 2011). It has generally been proposed that phylogenetic clustering patterns indicate that communities are structured based on environmental filters, since closely related species share similar traits and depend on similar ecological conditions (Webb et al., 2002). However, environmental filters can also cause phylogenetic overdispersion by trait convergence between distantly related species (Webb et al., 2002). In the case of crop fields, we detected patterns of phylogenetic clustering, which may be the result of environmental filters, probably favoring particular songs, as well as due to the reduction of available niches. It has been proposed that phylogenetic clustering as the result of environmental filters tend to be more important in open areas because species are likely to experience environmental adversities (Chazdon, 2003; Chazdon, 2008). Environmental adversity is even more important in open areas within semi-arid zones, such as our study site, since the conditions of temperature, solar radiation, and dryness are extreme, probably increasing the effect of environmental filters, and supporting the acoustic adaptation hypothesis (AAH).

On the other hand, a pattern of phylogenetic overdispersion also suggests competition by causing overdispersion of conserved traits (Webb et al., 2002). Therefore, it is likely that the negative relationship between phylogenetic structure and the bioacoustics index (BI), could be the result of competition for the acoustic space, because those closely related species that are expected to compete are not observed due to competitive exclusion. This mechanism would promote the diversification of acoustic signals (higher BI), resulting in sounds produced at different spectral (but not temporal) intervals supporting the acoustic niche hypothesis (Krause, 1993). In fact, it has been suggested that unlike the important effect of environmental filters in sites with adverse conditions, in sites where adversity is reduced, such as sites with greater vegetation cover and more conserved as the mesquiteras, biotic interactions such as competition become more important, showing patterns of phylogenetic overdispersion (Chazdon, 2003; Chazdon, 2008). However, to effectively test the role of competition for the acoustic space, we would need to specifically measure acoustic overlapping (or partition of the acoustic niche) within acoustic communities as an indirect estimation of competitive exclusion (Planqué & Slabbekoorn, 2008; Farina et al., 2011; Schmid, Römer & Riede, 2013; Krishnan & Tamma, 2016). It is important to mention that acoustic traits only explain a part of the variation. Community clustering or overdispersion could also be the result of environmental filters or competitive exclusion acting on other ecological traits related to resources such as food availability, nesting sites, protection sites against predators, or perhaps tolerance to solar radiation, among others.

In general, there are many examples in the literature that have shown the effect of environmental filters in avian communities along habitat perturbation gradients (e.g., Evans et al., 2018; Morelli et al., 2021) or elevational gradients (e.g., Ding et al., 2021; Montaño-Centellas, McCain & Loiselle, 2020; Montaño-Centellas, Loisell & Tingley, 2021). Most of these studies have shown that environmental conditions of urban or degraded habitats as well as elevation filter avian communities depending on their traits. Regarding acoustic communities, other studies have suggested community convergence by environmental filtering, such as acoustic adaptation to habitat characteristics (Planqué & Slabbekoorn, 2008; Cardoso & Price, 2010). Phylogenetic information is an important component in our understanding of acoustic community assembly and structure (Chhaya et al., 2021). However, there are few examples in the literature that have evaluated the role of environmental filters or competition from a phylogenetic perspective in avian acoustic communities. The few studies that have incorporated a phylogenetic approach have described patterns of overdispersion in some acoustic communities by divergence of spectral parameters of songs, as a result of competition for the acoustic space (Luther, 2009; Krishnan, 2019; García-Navas, Feliu & Blumstein, 2023). A different study showed that acoustic communities that occupy biomes with contrasting humidity conditions had a convergent overdispersed distribution in the acoustic signal space (Lahiri, Pathaw & Krishnan, 2021).

Conclusions

Overall, our data support the acoustic adaptation hypotheses, and suggest a role for environmental filters (in more open habitats) and probably competitive exclusion (in denser, more conserved habitats) in the bird community structure from the acoustic point of view. This study offers information about the influence of the habitat on the acoustic community structure which could be an approximation to explain the distribution of species from acoustic effects. This is one of the few studies that combines bioacoustics with tools offered by community ecology and community phylogenetics to understand how communities are structured from the acoustic point of view, and to understand how species respond to rapid anthropogenic change. The ecological niche theory has considered for many years different environmental variables as the dimensions that explain the distribution of a species (Hutchinson, 1957); however, the signal or acoustic space can also be considered as a dimension of the ecological niche, and therefore, a potential tool to better understand the species establishment and the structure of communities. Niche related processes such as competition or environmental filtering can be occurring in the soundscape, which are not visible, but are part of the behavior of the species and their establishment may depend on it. Future studies, besides directly quantifying acoustic overlapping in order to test for the acoustic niche hypothesis, should consider other important ecological traits and environmental variables to explain more precisely the role of the habitat in the coexistence of acoustic communities. It would be also interesting to include other plant associations in the region and compare the results with the winter season when several migratory birds stop in the area.

Supplemental Information

Supplemental Information 1 Shapiro-Wilk normality test values for phylogenetic indices (PD, NRI, NTI), acoustic indices (BI and ACI), vegetation structure, and excess attenuation

W = Shapiro-Wilk test statistic, P = probability value.

Supplemental Information 2 Raw data

The average measures for each variable for each plot used for analyses; raw data for each group of measures taken: vegetation measurements, excess attenuation, bioacoustic and phylogenetic indices.

We thank Javier Manjarrez, Victor Poblete and three anonymous reviewers for their comments and suggestions on a previous version of the manuscript; L. Morán Titla for field assistance; P. Miranda for his help in plant identification; J. Pacheco’s family for logistic support; A. Pacheco, J. Barragán Rivera, V. Mendoza and D. Pacheco for allowing us to work on their lands. This work constitutes partial fulfillment to the C. D. Morán-Titla’s masters in science (Maestría en Ecología Integrativa).

Additional Information and Declarations

Competing Interests

Author Contributions

Field Study Permissions

Data Availability

The authors declare there are no competing interests.

Christian D. Morán-Titla conceived and designed the experiments, performed the experiments, analyzed the data, prepared figures and/or tables, authored or reviewed drafts of the article, and approved the final draft.

Juan-Hector García-Chávez conceived and designed the experiments, analyzed the data, authored or reviewed drafts of the article, and approved the final draft.

Leonel Lopez-Toledo analyzed the data, authored or reviewed drafts of the article, and approved the final draft.

Clementina González conceived and designed the experiments, prepared figures and/or tables, authored or reviewed drafts of the article, and approved the final draft.

The following information was supplied relating to field study approvals (i.e., approving body and any reference numbers):

The authorities of the municipality and the Helia Bravo Hollis Botanical Garden verbally gave authorization to conduct our work at the Tehuacán-Cuicatlán Biosphere Reserve.

The following information was supplied regarding data availability:

The raw data is available in the Supplemental File.

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
