# Peer review of "Niche-related processes explain phylogenetic structure of acoustic bird communities in Mexico"

_PeerJ, doi:10.7717/peerj.18412_

## Round 0.1 · original submission · Major Revisions

Thank you very much for your manuscript titled “Niche-related processes explain phylogenetic structure of acoustic bird communities in Mexico” that you sent to PeerJ.

This study presents very valuable and relevant information about the Acoustic Adaptation Hypothesis. Authors measured and compared habitat density in three different habitats and examined sound attenuation levels between the habitats. Also, they examined how two acoustic indices varied among habitats as an indirect measure of competitive exclusion as it relates to the Acoustic Niche Hypothesis. Phylogenetic diversity and structure of the avian community were compared among habitats to examine whether species in the communities were more closely related than expected by chance.

As you will see below, comments from referee 1 suggest a minor revision while reviewer 2 suggests a major revision. Given this, I would like to see a major revision dealing with the comments. Their comments should provide a clear idea for you to review, hopefully improving the clarity and rigor of the presentation of your work. I will be happy to accept your article pending further revisions, detailed by the referees.

Reviewer 1 suggests renaming the acoustic indices and paying special attention to some points of the statistical analysis and the presentation of the references.
Reviewer 2 suggests improving the study hypothesis, in addition to considering other factors that may influence the phylogenetic structure between communities. They also make specific annotations within the pdf.

Please note that we consider these revisions to be important and your revised manuscript will likely need to be revised again.

·

Basic reporting

The paper addresses the problems of acoustic niche and acoustic adaptation in bird communities, and offers valuable information of the influences of the habitats on the acoustic structure of the birds. This information is proposed as a good approximation to explain the distribution of a specie. The work is of great interest. It is very well-written, with some minor observations:
- In Abstract, probably a typing error: "... les attenuation ..."

Experimental design

The main issues with this work are:
- To avoid confusion, a suggestion, perhaps would be better not to write that the authors used: "two bioacoustic indices", but simply to write that they used "two acoustic indices": 1) Bioacoustic Index (BI) and 2) Acoustic Complexity Index (ACI).
- On the other hand, the authors do not analyze the limitations that their work may have. In particular,
the PCA method. It has practical limitations particularly in the bio acoustic data analysis
based on non-linear features. Probably, the authors could review other techniques like unsupervised methods e.g., t-SNE and UMAP.

Validity of the findings

- The authors do not describe what future lines of their research could be done in order to
continue and improve this work.

Additional comments

- The authors should improve the style of their reference section. In particular, some of the journal
abbreviations do not have periods; and, some titles are capitalized at the beginning while others are not.

Reviewer 2 ·

Basic reporting

Generally well written, with suggested improvements to improve conciseness and especially clarity attached

Experimental design

Clarity to research questions could be improved- the three hypotheses are confusing and overlapping, as presented. Methods are adequate to address some hypotheses but see attached doc for suggestions to improve.

Validity of the findings

My sense is that authors should scale back somewhat on interpreting how results support hypotheses- again- see attachment

Annotated reviews are not available for download in order to protect the identity of reviewers who chose to remain anonymous.

---

## Round 0.2 · Minor Revisions

After reviewing this revised version of your manuscript, I see that you now have a more robust version because the main comments suggested by the two reviewers have been included. Now it is only necessary to make a few minor observations suggested by Reviewer 2 in order to have a final, publishable version of the manuscript.

·

Basic reporting

Dear Dr. Poblete,
As you will recall, you previously reviewed the manuscript noted below for PeerJ Life & Environment.

The authors have now revised the manuscript and I would be obliged if you could comment on the revision.

%%%%%%%%

20240917
Re-Reviewing Manuscript 100809v1

Title: Niche-related processes explain phylogenetic structure of acoustic bird communities in Mexico

Authors: Christian D. Morán-Titla, Juan García-Chávez, Leonel López-Toledo, Clementina González


Report:
In my opinion, it can be seen that the authors carefully reviewed the article.
They followed each of the suggestions and observations that were made.
The final version of the manuscript is a powerful contribution to the disciplinary field.
My congratulations to each of the authors for this valuable work done.

Experimental design

Dear Dr. Poblete,
As you will recall, you previously reviewed the manuscript noted below for PeerJ Life & Environment.

The authors have now revised the manuscript and I would be obliged if you could comment on the revision.

%%%%%%%%

20240917
Re-Reviewing Manuscript 100809v1

Title: Niche-related processes explain phylogenetic structure of acoustic bird communities in Mexico

Authors: Christian D. Morán-Titla, Juan García-Chávez, Leonel López-Toledo, Clementina González


Report:
In my opinion, it can be seen that the authors carefully reviewed the article.
They followed each of the suggestions and observations that were made.
The final version of the manuscript is a powerful contribution to the disciplinary field.
My congratulations to each of the authors for this valuable work done.

Validity of the findings

Dear Dr. Poblete,
As you will recall, you previously reviewed the manuscript noted below for PeerJ Life & Environment.

The authors have now revised the manuscript and I would be obliged if you could comment on the revision.

%%%%%%%%

20240917
Re-Reviewing Manuscript 100809v1

Title: Niche-related processes explain phylogenetic structure of acoustic bird communities in Mexico

Authors: Christian D. Morán-Titla, Juan García-Chávez, Leonel López-Toledo, Clementina González


Report:
In my opinion, it can be seen that the authors carefully reviewed the article.
They followed each of the suggestions and observations that were made.
The final version of the manuscript is a powerful contribution to the disciplinary field.
My congratulations to each of the authors for this valuable work done.

Additional comments

Dear Dr. Poblete,
As you will recall, you previously reviewed the manuscript noted below for PeerJ Life & Environment.

The authors have now revised the manuscript and I would be obliged if you could comment on the revision.

%%%%%%%%

20240917
Re-Reviewing Manuscript 100809v1

Title: Niche-related processes explain phylogenetic structure of acoustic bird communities in Mexico

Authors: Christian D. Morán-Titla, Juan García-Chávez, Leonel López-Toledo, Clementina González


Report:
In my opinion, it can be seen that the authors carefully reviewed the article.
They followed each of the suggestions and observations that were made.
The final version of the manuscript is a powerful contribution to the disciplinary field.
My congratulations to each of the authors for this valuable work done.

Reviewer 2 ·

Basic reporting

Authors sufficiently addressed my primary concerns from the earlier version of this manuscript. I have attached a summary of some suggested additional minor edits

Experimental design

Authors removed the portion of the manuscript that dealt with testing the Acoustic Niche Hypothesis. This greatly improved experimental design

Validity of the findings

Findings currently appear to be valid

Additional comments

Authors sufficiently addressed my primary concerns from the earlier version of this manuscript. I have attached a summary of some suggested additional minor edits

Annotated reviews are not available for download in order to protect the identity of reviewers who chose to remain anonymous.

---

## Round 0.3 · accepted · Accept

After reviewing this revised version of your manuscript, I see that comments suggested by the reviewers have been included. Therefore, I am satisfied with the current version and consider it ready for publication.